# The Etiology of Pancreatic Manifestations in Patients with Inflammatory Bowel Disease

**DOI:** 10.3390/jcm8070916

**Published:** 2019-06-26

**Authors:** Tomoya Iida, Kohei Wagatsuma, Daisuke Hirayama, Yoshihiro Yokoyama, Hiroshi Nakase

**Affiliations:** Department of Gastroenterology and Hepatology, Sapporo Medical University School of Medicine, Sapporo 060-8543, Japan

**Keywords:** pancreatitis, autoimmune pancreatitis, inflammatory bowel disease

## Abstract

Inflammatory bowel disease (IBD) is an idiopathic chronic and recurrent condition that comprises Crohn’s disease and ulcerative colitis. A pancreatic lesion is one of the extraintestinal lesions in patients with IBD. Acute pancreatitis is the representative manifestation, and various causes of pancreatitis have been reported, including those involving adverse effects of drug therapies such as 5-aminosalicylic acid and thiopurines, gall stones, gastrointestinal lesions on the duodenum, iatrogenic harm accompanying endoscopic procedures such as balloon endoscopy, and autoimmunity. Of these potential causes, autoimmune pancreatitis (AIP) is a relatively newly recognized disease and is being increasingly diagnosed in IBD. AIP cases can be divided into type 1 cases involving lymphocytes and IgG4-positive plasma cells, and type 2 cases primarily involving neutrophils; the majority of AIP cases complicating IBD are type 2. The association between IBD and chronic pancreatitis, exocrine pancreatic insufficiency, pancreatic cancer, etc. has also been suggested; however, studies with high-quality level evidence are limited, and much remains unknown. In this review, we provide an overview of the etiology of pancreatic manifestation in patients with IBD.

## 1. Introduction

Inflammatory bowel disease (IBD) is an idiopathic chronic and recurrent condition that comprises Crohn’s disease (CD) and ulcerative colitis (UC), and the number of affected patients has risen sharply in recent years in Europe, the United States of America, and Japan [1]. Genetic predisposition (innate and acquired immunity, cytokine, and racial difference) and environmental factors (diet, drugs, smoking, and infection) are greatly involved in the onset of IBD, and intestinal immune abnormalities are caused by the involvement of the state of dysbiosis, which is believed to cause IBD [2]. IBD is a systemic disease that often manifests extraintestinally and has an incidence rate of 6–47% [3]. Typical extraintestinal manifestations of IBD include dermatologic, musculoskeletal, ocular, oral, pulmonary, hepatobiliary, and pancreatic lesions [4,5,6,7,8].

Among them, the majority of pancreatic manifestations accompanied by IBD are increased pancreatic enzyme levels and pancreatitis. In recent years, studies have been conducted on their various pathological conditions. In this review, we provide an overview of the etiology and treatment of pancreatitis in patients with IBD.

## 2. Pancreatic Manifestations Accompanied by IBD

With regard to pancreatic manifestations accompanied by IBD, in 1950, Ball et al. [9] reported on pancreatic manifestations accompanied by UC observed in autopsies for the first time, and in 1956, in an investigation also involving autopsies, Chapin et al. [10] reported that histological changes in the pancreas associated with regional enteritis were primarily interlobar and periductal fibrosis and swelling of the acinar cells. In addition, in 1967, Frey reported on acute pancreatitis as a complication of UC [11]. Subsequently, studies on IBD, particularly pancreatitis accompanied by CD, have been gradually reported since the 1970s in Europe and the United States [12,13,14].

Currently, suggested etiologies of increased levels of pancreatic enzymes or pancreatitis in IBD include immune abnormalities, genetic predisposition, microcirculation disorders, malnutrition, dehydration, enterobacteria with the ability to produce amylase, and the excretion of enteric amylase into the bloodstream. Immune abnormalities are involved in the pathogenesis of IBD, and Stocker et al. [15] reported that autoantibodies against pancreatic juice were found in the blood of 39% and 4% of patients with CD and UC, respectively. Borkje et al. [16] reported four cases of complicating UC, sclerosing cholangitis, and chronic pancreatitis and discussed pancreatic manifestations in IBD and genetic predispositions. However, regarding the relationship between increased levels of pancreatic enzymes and disease activity, severity, and extent of involvement of IBD, a study reported a correlation with the extent of the manifestation and histologic activity of IBD [17], whereas another study reported no correlation with changes in serum amylase levels or disease progression of CD [18]. 

In addition, pathological pancreatic manifestations accompanied by IBD have been shown to include the asymptomatic elevation of pancreatic enzymes, acute pancreatitis (AP), chronic pancreatitis (CP), exocrine pancreatic insufficiency (EPI), and pancreatic cancer (PC) (Table 1). Although these pancreatic manifestations are mutually associated and are not formed by a simple pathological condition, representative reports on pancreatic manifestations associated with IBD are summarized below.

### 2.1. Asymptomatic Elevation of Pancreatic Enzymes

Due to the factors mentioned above, patients with IBD may show elevated pancreatic enzymes. Amylase and lipase are common pancreatic enzymes. Amylase is divided into S-amylase and P-amylase, which are secreted from the salivary glands and pancreas, respectively. It should be noted that lipase is more specific to the pancreas than amylase. Heikius et al. [17] examined 237 patients with IBD and prospectively investigated the association between elevated levels of pancreatic enzymes and endoscopic and pathological findings. The percentages of hyperamylasemia and hyperlipasemia among all patients were 11% and 7%, respectively, whereas they were 17% and 9% among patients with CD, 9% and 7% among patients with UC, and 10% and 5% among patients with indeterminate colitis, respectively. Furthermore, they reported that amylase and P-amylase levels were correlated with endoscopic and pathological severity. Katz et al. [19] examined the asymptomatic elevation of pancreatic enzymes in 180 patients with IBD (UC: 83, CD: 97), and reported that amylase elevation was observed in eight (8.2%) patients with CD, of which four (4.1%) showed P-amylase elevation, but amylase elevation was not associated with the location of the manifestation or disease progression. In addition, Bokemeyer [20] conducted a prospective trial on 136 patients with IBD (UC: 70, CD: 66) and reported that 19 patients (14%) showed amylase or lipase elevation without abdominal symptoms, and amylase or lipase levels were not correlated with Crohn’s Disease Activity Index (CDAI), Clinical Activity Index (CAI), or C-reactive protein (CRP) levels. These results indicate that the significance of the asymptomatic elevation of pancreatic enzymes in patients with IBD remains open to discussion.

### 2.2. Acute Pancreatitis 

AP is an acute sterile inflammation wherein the pancreas undergoes autolysis due to the activation of pancreatic enzymes in the pancreas for various reasons. In the early stages, patients suffer from bouts of acute abdominal pain and tenderness in the upper abdomen. It is characterized by varying degrees of severity, ranging from mild cases where inflammation remains in the pancreas and improves in a few days to severe cases where inflammation spreads to the entire body, which becomes life-threatening. In severe cases, mortality is as high as 10%, and approximately half of patients die because of impaired circulation in the early stages, whereas others die because of a complicating infection in the late stages [21]. Blood tests and contrast-enhanced computed tomography (CT) are essential to the diagnosis of AP. In terms of treatment, patients undergo adequate fluid replacement and monitoring. Patients are administered antibiotics, depending on the severity, and they are treated for local complications [22]. The endoscopic removal of gallstones should be considered if they are causing AP, and the administration of drugs must immediately be discontinued if involvement of drugs is suspected.

AP seems to be a complication of IBD in some cases, and it may prove fatal. Factors contributing to the development of AP as a complication of IBD include idiopathic disease, drugs, gallstones, duodenal papillary lesions, procedural accidents due to endoscopic balloons, primary sclerosing cholangitis (PSC), and autoimmune pancreatitis (AIP). In the West, the incidence of acute pancreatitis in IBD is reported to be higher in CD than UC [8]. The percentage varies by study, but the odd rate for AP seems to be as high as 4.3 times (3.1%) and 2.1 times (1.2%) in CD and UC, respectively, compared to the general population [23]. One study, including patients with CD after a 10-year follow-up, showed a significantly higher incidence of AP in CD patients than in the general population (1.4% vs. 0.007%, respectively) [17]. A population-based cohort study in Taiwan showed that the overall incidence of AP was 3.56-fold higher in patients with IBD (31.8 per 10,000 person-years) in comparison with those without IBD (8.91 per 10,000 person-years) [24]. Furthermore, in a recent retrospective study involving 602 patients with IBD (UC: 57.5%, CD: 42.5%), four patients (0.6%) developed AP during an average period of 5.8 years [25]. However, some patients may possibly have been overdiagnosed because, as mentioned above, the elevation of pancreatic enzymes is highly frequent among patients with IBD, and differentiating between abdominal pain caused by IBD and abdominal pain due to pancreatitis is difficult at times. When AP is suspected, the physician must closely observe the patient and always perform imaging tests to make an accurate diagnosis. Different pathological conditions that present AP are described in detail below.

### 2.3. Chronic Pancreatitis

CP is a progressive chronic inflammatory fibrotic disease with pathological features, such as decreased exocrine and endocrine function due to inflammation, including irregular fibrosis, cellular infiltration, parenchymal loss, and granulation tissue inside the pancreas [26]. Smoking, drinking, family history, etc. are involved in the development of CP, and a recent review reported that the annual global incidence of CP was 10 cases [27]. The mortality of patients with CP is approximately twice as high as that of the general population, and a recent meta-analysis also showed that CP was an obvious risk factor for pancreatic cancer [28]. Recurrent upper abdominal and back pain often appear as initial symptoms of CP, and abdominal pain attenuates as the disease stage progresses, and the endocrine and exocrine function of the pancreas gradually decreases [26]. Close observation of clinical signs; diagnostic imaging, including abdominal ultrasonography, endoscopic ultrasonography, and CT and magnetic resonance imaging (MRI); and endocrine/exocrine function tests are important for staging of the disease [29]. Although treatment varies depending on the disease stage, it can largely be divided into medical and surgical treatments. Medical treatments include lifestyle guidance involving smoking cessation and alcohol abstinence [30], diet and nutrition therapy [31], stepwise drug therapy [32,33], and endoscopic treatment of pancreatic calculi [34]. Surgical treatments include pancreaticoduodenectomy, pylorus-preserving pancreaticoduodenectomy, and duodenum-preserving pancreatic head resection, including Beger or Frey procedures. A recent meta-analysis showed that all of these procedures were effective [35]. Therefore, combination therapy based on the disease stage is required for the treatment of CP along with an accurate diagnosis of the disease stage. 

There are a limited number of reports on CP that has manifested in patients with IBD. Barthet et al. [36] retrospectively examined 20 idiopathic patients with CP (UC: 6, CD: 14) who developed IBD. Hyperamylasemia was found in only 44% and 64% of patients with UC and CD, respectively, indicating low sensitivity. Furthermore, pancreatitis preceded UC in 58% of UC patients, whereas 56% of patients with CD developed pancreatitis after being diagnosed with CD. In UC, CP is associated with the pancolitis type and total colectomy, suggesting an association with disease progression. Moreover, a recent nationwide population-based cohort study in Taiwan showed that the incidence of CP in patients with IBD was 10.3 times (5.75 vs. 0.56 per 10,000 person-year) higher than that in non-IBD patients, whereas, conversely, the risk of developing IBD was significantly higher in patients with CP (CD: adjusted hazard ratio [aHR]  =  12.9, 95% confidence interval [CI]  =  5.15–32.5, CD: aHR  =  2.80, 95% CI  =  1.00–7.86) [37]. CP also develops in children with IBD, and a case has been reported wherein CP developed as a precursor manifestation before the diagnosis of CD [38]. Therefore, an association between CP and IBD has been presumed. However, there are only a few reports, and further investigation is required.

### 2.4. Exocrine Pancreatic Insufficiency

EPI is a generic term that covers pathological conditions characterized by the digestive malabsorption of fats, proteins, and carbohydrates due to an insufficiency of pancreatic enzymes that are normally secreted by the pancreas. EPI causes steatorrhea, diarrhea, fat-soluble vitamin deficiency, essential fatty acid deficiency, etc., eventually leading to malnutrition and weight loss [39]. EPI is commonly caused by diseases that destroy the pancreatic parenchyma, such as chronic pancreatitis and pancreatic cancer [40]. IBD is a disease that causes EPI, which has been reported in recent studies. Possible mechanisms for the development of EPI in CD include pancreatic autoantibodies, duodenal reflux, and reduced hormone secretion [39]. The incidence of EPI based on low fecal elastase levels varies between 14% and 30% of patients with CD [41,42]. Conversely, in UC, it was reported that 22% of patients had fecal elastase levels ≤200 μg/g, and 9% had severe EPI (fecal elastase ≤100 μg/g) [41]. In addition, using a secretin-cerulein test, 50% of patients with UC demonstrated bicarbonate and/or enzyme insufficiency, while 74% had an abnormal para-aminobenzoic acid (PABA) test [43,44]. A number of reports of EPI associated with IBD have been published to date. However, the number of those with high evidence levels is limited, and further examination is required.

### 2.5. Pancreatic Cancer

Due to chronic inflammation, there is concern that gastrointestinal cancer may develop in patients with chronic IBD. However, a recent meta-analysis conducted in the West reported that the risk of developing colorectal cancer for patients with IBD was only 1.7 times higher than that for non-IBD patients [45], showing a declining trend compared with previous studies. By contrast, the percentage of extraintestinal cancer associated with IBD has been reported to increase over time. Malignant lymphoma [46,47] and melanoma [48] are typical examples.

A cohort study on pancreatic lesions has been conducted in relation to PSC. It was reported that 224 (2%) of 11,028 patients with IBD (UC: 5,522, CD: 5,506) had developed PSC, and the incidence of PC was higher, with an odds ratio (OR) of 11.22 (95% CI: 4.11–30.62), in patients with IBD with complicating PSC than in patients with IBD without complicating PSC [49]. Furthermore, the results of a nationwide population-based study on IBD and carcinogenesis conducted in Asia were recently reported. Of the 15,291 patients with IBD (UC: 9,785, CD: 5,506), 273 had cancer (1.8%) (UC: 198, CD: 75). The standardized incidence ratio of PC in men was 2.42 (95% CI: 0.66–6.21) and 5.59 (95% CI: 0.68–20.20) for UC and CD, respectively, whereas that in women was 1.26 (95% CI: 0.03–7.02) and 8.58 (95% CI: 1.04–31.00) for UC and CD, respectively [50]. Based on these reports, there seemed to be an association between IBD and PC, although racial and sex differences were observed. One of the reasons for this is that immunosuppressive therapy like the use of thiopurines has been associated with an increased risk of cancers owing to impaired immune surveillance. Second, chronic inflammation in PSC and IBD and associated immune activation has been associated with an increased risk of cancers, independent of the effect of immunosuppressive therapy, as has also been observed in other chronic inflammatory diseases [49]. 

Similar to IBD without complicating PC, image evaluation, including contrast-enhanced CT and magnetic resonance cholangiopancreatography (MRCP), is important for diagnosis. Both CT and MRI are highly sensitive in the detection of pancreatic cancer, with up to 96% and 93.5% sensitivity, respectively [51]. A definitive pathologic diagnosis is made using endoscopic retrograde cholangiopancreatography (ERCP) and endoscopic ultrasound-guided fine needle aspiration (EUS-FNA) [52]. Depending on the disease state, surgical resection and chemotherapy are selected for treatment. However, patients with IBD should preferably undergo regular screening tests for extraintestinal manifestations, including pancreatic manifestations [53].

## 3. AP Accompanied by IBD

### 3.1. Idiopathic

As described above, patients with IBD develop AP due to various causes, and AP resulting from an unknown cause is referred to as idiopathic pancreatitis (IP). In a retrospective study that examined 1057 patients with IBD, the incidence of IP was 0.38% (4/1057), whereas the incidence of pancreatitis was 2.74% (29/1057) [54]. Furthermore, in a retrospective study that examined 48 patients with CD who developed pancreatitis, IP only accounted for 8% of the causes leading to pancreatitis [55]. In Seyrig’s study, IP was only 1.5% (five of 331 patients with IBD) [13]. In Heikius’s study, the incidence of IP was 3% and 4% in patients with IBD and a subgroup with CD, respectively [17]. In a study that also included children, 0.06% (2/3,500) and 2.17% (10/460) of adults and children developed IP before being diagnosed with IBD, respectively, and the authors discuss that proper follow-up is required for children in particular after they develop IP [56].

Although the diagnosis/treatment of IP is similar to general AP [22], the various factors described below must be excluded. Accurate medical interviews, blood tests, and imaging tests are indispensable.

### 3.2. Drugs

Pancreatitis caused by drugs administered for therapeutic purposes is called drug-induced pancreatitis (DIP), and >100 drugs cause it. Clinically, DIP manifests as acute pancreatitis, and it does not progress to chronic pancreatitis [57]. Most cases are mild with a positive prognosis, but caution is required when it becomes severe as it can lead to death in some cases [58,59]. There is no sex difference or a peak age of onset in DIP. The predilection period of DIP is related to the pathogenic mechanism of pancreatitis. Although pancreatitis due to drug-specific toxicity develops in a short period (within 24 h), there are very few clinical cases, and allergic reactions to drugs are involved in many cases of DIP. In such cases, the predilection period of pancreatitis is a few days after administration, and most of the patients develop pancreatitis within one month [57,60]. Blomgren et al. [61] examined risk factors in patients who were prone to developing DIP and found that a previous history of digestive diseases (OR: 1.5 [1.1–1.9]) and IBD (OR: 3.4 [1.5–7.9]) were among these risk factors, suggesting a relationship between IBD and DIP.

Among the therapeutic drugs for IBD, mesalazine [62,63] and thiopurine drugs [64,65] are typical examples of drugs that cause DIP. Both peroral and transanal administration of mesalazine can cause SIP. In a prospective study that examined the onset of pancreatitis in 510 patients with IBD (UC: 157, CD: 338, indeterminate colitis: 15) on azathioprine, 37 (7.3%) developed pancreatitis, and a relationship was found between the onset of pancreatitis and smoking [66]. Bermejo et al. [67] reported that the risk of developing DIP due to azathioprine was higher in women than in men, and higher in those with CD than with UC. Wilson et al. [68] conducted a retrospective study on 373 patients with IBD who were administered azathioprine and reported that HLA polymorphism was a major predictor for DIP caused by azathioprine. Furthermore, the use of biologics by patients with IBD is becoming more frequent, and there have been occasional reports of pancreatitis that developed due to biologics in recent years [69,70].

Diagnosis of DIP is based on (1) the onset of pancreatitis during the administration of the drug, (2) remission of pancreatitis upon drug discontinuation, and (3) relapse upon re-administration [61]. However, because re-administration poses a danger and is ethically problematic, it is not performed at present. Drug lymphocyte stimulation tests (DLSTs) also remain auxiliary when making a diagnosis, meaning that the diagnosis of DIP is difficult. In a previous study, Badalov et al. [57] classified drugs that may cause DIP into five classes (classes Ia, Ib, II, III, and IV in descending order of likelihood), and we believe it holds some referential value. In this study, mesalazine is classified as class Ia, whereas thiopurine drugs are classified as class Ib.

The mainstay of treatment is the immediate discontinuation of the culprit drug, and subsequent treatment is similar to that for idiopathic AP [22]. Being unable to use mesalazine and thiopurine drugs, which are basic treatment drugs for IBD, affects the patient’s prognosis, but caution is required because acute pancreatitis often recurs upon re-administration of the drug in DIP. Regular amylase measurements should be performed, and attention should be paid to abdominal symptoms one month after starting the administration of a new drug. Additionally, tests, discontinuation of the drug, and treatment should be performed while keeping DIP in mind if the patient develops AP.

### 3.3. Gall Stones

Cholelithiasis exhibits choledocholithiasis and causes AP [71]. In a recent meta-analysis that examined AP in 2,341,007 subjects in 36 countries, it was reported that cholelithiasis accounted for as much as 42% of AP causes, and that the number was particularly high in Latin America [72]. In IBD, CD is reported to show a high incidence of complicating cholelithiasis. Gallstone formation in CD is considered to be due to ileal manifestations, ileal resection, and long-term parenteral nutrition. The prevalence of cholelithiasis in CD patients ranges from 11% to 34%, whereas it ranges from 5.5% to 15% in non-IBD patients [73]. Zhang et al reported that the risk of gallstones in CD was increased by an odds ratio (OR) of 2.05 in a meta-analysis [74]. Conversely, the incidence of complicating cholelithiasis in UC remains controversial, and although a meta-analysis reported that the incidence of cholelithiasis did not increase in UC [74], a recent retrospective study conducted in Asia reported that the incidence of cholelithiasis was 3.9% (24/622) and 8% (25/311) in the control group and patients with UC, respectively, with an OR of 2.18, indicating that patients with UC had a significantly higher chance of developing complicating cholelithiasis [75]. In a recent nationwide population-based cohort study conducted in Asia, although the likelihood of developing cholecystolithiasis and choledocholithiasis or hepaticolithiasis was significantly higher than the control group, with an OR of 1.76 (95% CI: 1.34–2.61) and 2.78 (95% CI = 1.18–6.51), respectively, biliary lithiasis did not increase in patients with UC [76]. Furthermore, the risk factors for developing complicating cholelithiasis in IBD were reported in CD, age at diagnosis, disease activity and duration, non-steroidal anti-inflammatory drug (NSAID) intake, extra-intestinal manifestations, and intestinal surgery [77].

In terms of diagnosing AP caused by cholelithiasis, regardless of whether there is complicating IBD, cholelithiasis MRCP and endoscopic ultrasound (EUS) are effective for diagnosing choledocholithiasis, whereas contrast-enhanced CT is effective for diagnosing and determining the severity of AP. Endoscopic lithotrity is the first-line treatment [78], and acute-phase treatment after lithotrity is similar to that for IP [22]. In general, cholecystectomy is recommended for choledocholithiasis in addition to the treatment of choledocholithiasis. In a retrospective study that examined 4516 patients who underwent ERCP, the risk of subsequent recurrent biliary events was reported to decrease by 87% (*p* < 0.001) and 88% (*p* < 0.001) in the early and late cholecystectomy groups, respectively, compared with the non-cholecystectomy group [79]. In the case of limiting gallstone pancreatitis, it was reported that both cholecystectomy and ES are superior to conservative treatment in reducing the incidence of recurrent attacks of acute biliary pancreatitis [80]. Furthermore, cholecystectomy with endoscopic sphincterotomy has been reported to significantly decrease the recurrence of gallstone pancreatitis compared with endoscopic sphincterotomy alone (OR 0.35, 95% CI 0.24–0.49, *p* < 0.001) [81].

### 3.4. Gastrointestinal Lesions on the Duodenum

With regard to duodenal manifestations associated with CD, Legge et al. [12] reported that three of 10 patients with CD and manifestations in the duodenal papilla, such as fistulas, developed reflux pancreatitis. Furthermore, Meltzer et al. [82] reported that the development of AP should be considered in Crohn’s disease, wherein stenosis of the duodenum occurs and duodenopancreatic reflux is caused. Gschwantler et al. [14] reported cases wherein pancreatic manifestations in patients with CD directly infiltrated the pancreas. Therefore, although there are only a few coherent studies, we must pay attention to the development of AP in patients with CD with duodenal manifestations.

### 3.5. Endoscopic Procedures

Small bowel enteroscopy (BE) is a useful examination for the diagnosis/treatment of CD [83], and it is widely used. BE comprises single-balloon enteroscopy (SBE) and double-balloon enteroscopy (DBE), each has a peroral and transanal approach, and procedural accidents may occur in both [84,85,86]. One of the procedural accidents is AP, and almost all cases of post-BE AP, occurs after peroral BE, with pancreatitis developing primarily in the pancreatic tail in most cases. AP has been reported to develop after a patient underwent transanal BE [87], but it is assumed to be due to the physical load on the duodenum and pancreas resulting from excessive shortening procedures and the long examination time. In an animal-model study in which pigs were used, impaired blood flow in the direction toward the pancreatic tail has been suggested to be involved in the development of AP [88].

Post-BE hyperamylasemia without pancreatitis was observed in 23 (17%) of the patients in a prospective trial with 135 patients who underwent peroral DBE [89], and in 36 (39%) of the patients in a similar prospective trial with 92 patients who underwent peroral DBE, wherein an examination time of ≥60 min was a risk factor for hyperamylasemia [90]. Furthermore, a small-scale trial showed that nine (75%) of 12 patients who underwent DBE manifested hyperamylasemia without pancreatitis [91]. By contrast, with regard to post-BE pancreatitis, the two prospective trials mentioned above showed that the incidence of post-BE AP was 0.7% and 3.2% [89,90]. However, in a prospective trial involving 48 patients who underwent peroral DBE, the incidence of post-BE AP was reported to be high at 12.5%, and the total insertion length, duration, and duration between the first and second inflations of the balloon were risk factors of developing AP [92]. Similar to post-BE hyperamylasemia, there is much inter-trial variation. This may be due to varying degrees of workmanship of the technicians as the incidence is correlated with the examination time and because the incidence is higher in studies with fewer cases.

With regard to the difference in procedural accidents between SBE and DBE, a randomized multicenter trial (SBE: 65, DBE: 65) with 130 subjects showed no difference [84]. Furthermore, meta-analyses of patients who underwent BE concluded that no difference was found in procedural accidents between SBE and DBE [85,86]. In the case of post-BE AP, treatment similar to that for general AP is provided. However, we must acknowledge that the patient may present with AP after BP and consider performing examinations in as short a period as possible to prevent the onset of AP.

### 3.6. Primary Sclerosing Cholangitis 

PSC is an intractable disease that exhibits multiple diffuse stenoses in the intrahepatic/extrahepatic bile ducts that cause cholestasis [93]. The incidence and prevalence rates of PSC are high in North America and northern Europe, with a reported incidence of 0.4–1.22 and a prevalence of 4.15–16.2 [94]. Genetic factors and enteric environments are reported to be involved in its onset. A study conducted in the West showed that 50–80% of patients with PSC developed complicating IBD, and common disease susceptibility genes with IBD were found [95]. Genetically, PSC is known to be more strongly associated with UC than CD, and patients present with atypical findings, such as right-side dominance and a lack of rectal manifestations [96]. 

It has been reported that, although its incidence is low, PSC patients may develop pancreatitis [97,98]. The acute pancreatitis was probably due to the reflux of bile and sludge into the pancreatic duct due to stricture of the distal part of common bile and pancreatic ducts. Therefore, endoscopic biliary stent placement was reported to be effective [97]. However, the mechanism by which PSC patients develop pancreatitis remains unknown. Furthermore, PSC has been reported to be an independent factor for post-ERCP pancreatitis [99], and particular care is required when examining patients with PSC with complicating IBD.

### 3.7. Autoimmune Pancreatitis

AIP is a rare yet increasing autoimmune disease. In the international consensus diagnostic criteria (ICDC), AIP is classified into types 1 and 2 from a clinicopathological perspective [100]. The involvement of autoimmune mechanisms is suspected in type 1 AIP as the patient presents with hypergammaglobulinemia; elevated serum IgG and IgG4 levels; tests positive for antinuclear antibodies and autoantibodies, such as the rheumatoid factor; and responsiveness to steroid treatment. There is a high possibility that it is a pancreatic manifestation of an IgG4-related disease. It is common in Asia, with a mean age of onset of 62 years. Men are more prone (74%) to develop this disease, and jaundice develops in many cases. Conversely, type 2 AIP is common in the West, with a low mean age of onset of 45 years. Some reports state no sex difference, whereas others claim that men are slightly more prone to develop the disease. The patients develop AP in many cases. Type 2 AIP has a high incidence of developing IBD, particularly UC, whereas it presents few serological abnormalities, such as low IgG4 levels [101,102,103]. The ratio of types 1 and 2 in AIP varies by population. Although an international study showed 86 cases (8%) of type 2 AIP of 1064 cases of AIP, the incidence is even lower in Asia. Some studies showed that type 2 AIP accounts for approximately 10–40% of cases in Asia [101,104,105,106]. 

In dynamic CT findings, diffuse parenchymal enlargement with delayed enhancement is one of the characteristic radiologic features of AIP. Enlargement accompanied by effacement of the lobular contour of the pancreas gives the gland a featureless or sausage-shaped appearance. A low-attenuating capsule-like rim around the enlarged pancreas is also a relatively specific finding for AIP [103]. In other cases, AIP may present focal changes and differential diagnosis from PC may be required. In such cases, the use of PET/CT is reported to be effective in differentiating between the two [107]. With regard to the pancreatic duct, AIP is characterized by diffuse or focal stenosis of the main pancreatic duct in ERP, and biliary stenosis, renal lesions, and retroperitoneal fibrosis are findings that support the diagnosis of AIP [103]. With regard to different CT findings between types 1 and 2, a study conducted in South Korea showed that no significant difference was found between types 1 and 2 in the incidence of diffuse pancreatic enlargement (62% vs. 73%, respectively) and lower biliary stricture (74% vs. 57%, respectively) [104]. By contrast, a study conducted in Japan showed that the incidence of diffuse pancreatic enlargement in type 2 AIP associated with IBD was similar to that of previous studies (71%), but, compared with type 2 AIP, the incidence of pancreatic head enlargement (53% vs. 75%, respectively) and lower biliary stricture (13% vs. 76%, respectively) was significantly low in type 1 AIP [105].

Pathologically, type 1 AIP presents lymphoplasmacytic sclerosing pancreatitis (LPSP), which is characterized by marked lymphocyte and IgG4-positive plasma cell infiltration, storiform fibrosis, and obstructive phlebitis. In contrast, type 2 AIP is characterized by a granulocyte epithelial lesion (GEL) caused by neutrophil infiltration [108,109]. Table 2 shows the clinicopathological characteristics of types 1 and 2 AIP.

The prevalence of IBD in patients with AIP has been reported to be higher than that in the general population [110]. As described above, the relationship between IBD and AIP mainly involves UC and type 2 AIP. Specifically, the rate of UC in patients with AIP is up to 35% [111,112]. On the contrary, the incidence of AIP in patients with IBD is low. A Japanese study conducted on 1,751 patients with IBD (UC: 961, CD: 790) showed a 0.4% (7 (UC: 5, CD: 2)/1,751) prevalence of type 2 AIP [106]. The impact of AIP in the natural history of IBD has not been cleared. UC seemed to be more severe, with four out of 10 patients requiring colectomy [110,113]. On the contrary, Ueki et al. [105] found no differences in disease extent or activity in UC patients with or without AIP.

A comprehensive diagnosis of AIP is made based on imaging findings, including the enlargement of the pancreas and stricture of the main pancreatic duct; serum IgG4 levels; pathological findings; extrapancreatic manifestations; and responsiveness to steroids. A definitive diagnosis of type 2 AIP, which is strongly associated with IBD, is made if a GEL is confirmed in the pancreas biopsy specimen or resected specimen, that is, if level 1 primary findings (histological images of the pancreas) of type 2 AIP are made. When AIP is suspected in imaging and level 2 findings (infiltration of neutrophils, lymphocytes, and plasma cells into the acini is noted with no IgG4-positive cells or very few at 0–10 cells/HPF) of type 2 AIP are made histologically, a definitive diagnosis of type 2 AIP is made, even if there is complicating IBD, and the patient responds to steroid treatment [100].

Treatment of type 1 AIP differs for the remission induction period, remission maintenance period, and relapse. During the remission induction period, basic treatment comprises systemic administration (0.6 mg/kg/day or 40 mg/day) of prednisolone (PSL), which is gradually reduced in increments of 5 mg after four weeks of administration [103]. In a recent study, methotrexate was reported to be effective for inducing remission [114]. Although there is room for discussion regarding remission maintenance therapy, in actual clinical settings, patients have to be administered with small doses of PSL (2.5–10 mg/day) for a prolonged period in many cases. In a recent randomized controlled trial (RCT) that compared the group in which steroids were gradually discontinued after remission was induced and the group in which PSL was continued for three years at a dose of 5–7.5 mg/day, relapse-free survival was reported to be significantly lower in the latter group (*p* = 0.011) [115]. Although a study reported that, similar to IBD treatment, the use of immunomodulators was also effective for reducing the dose of steroids during the remission maintenance period [116], much remains unknown because no inter-drug comparison has been conducted. Relapse has been reported to occur in approximately 60% of patients [117,118], and reintroducing the systemic administration of PSL and using immunomodulators in combination are considered. Rituximab has been reported to be effective [116]. Similar to type 1 AIP, the systemic administration of PSL (0.6 mg/kg/day or 40 mg/day) is performed for type 2 AIP, and the dose is gradually reduced. The relapse rate of type 2 AIP is lower than that of type 1 AIP, and the relapse rate at 12 months was 25% when the patient did not undergo maintenance therapy after remission induction [119]. As such, the evidence level for maintenance therapy is low, and it is not recommended. Furthermore, in type 2 AIP with complicating IBD, the initial administration of drugs for IBD may be an option, and administering steroids to patients with non-alleviating pancreatitis can be considered because IBD tends to be in its active phase. 

## 4. Conclusions

Pancreatic manifestations associated with IBD are one set of extraintestinal manifestations, and controlling their progression is extremely important because poor control also affects the treatment of intestinal lesions. AP is a typical example, and it is caused by various factors, including idiopathic, drug-induced, and AIP cases. Furthermore, much remains unknown regarding the relationship of IBD to CP, EPI, and pancreatic cancer, because only few studies have presented high-quality level evidence. We believe future research will contribute to enhancing the quality of life of patients with IBD by elucidating its mechanism as the etiology of pancreatic manifestations in patients with IBD becomes clear.

## Figures and Tables

**Table 1 jcm-08-00916-t001:** Pancreatic manifestations accompanied by inflammatory bowel disease.

1. Asymptomatic elevation of pancreatic enzyme
2. Acute pancreatitis (AP)
a. Idiopathic
b. Drugs
c. Gall stones
d. Gastrointestinal lesions on the duodenum
e. Endoscopic procedures
f. Primary sclerosing cholangitis (PSC)
g. Autoimmune pancreatitis (AIP)
3. Chronic pancreatitis (CP)
4. Exocrine pancreatic insufficiency (EPI)
5. Pancreatic cancer (PC)

**Table 2 jcm-08-00916-t002:** Clinicopathological characteristics in type 1 and 2 autoimmune pancreatitis.

	Type 1	Type 2
Racial difference	Asian > European/American	Asian < European/American
Peak age of onset	60s	40s
Sex difference	Mainly men	None or more common in men
Mode of onset	Jaundice is predominant	Acute pancreatitis is predominant
IgG/IgG4	Elevated	Normal
Autoantibody positive	High frequency	Low frequency
Diffuse pancreatic enlargement	High frequency	High frequency
Lower biliary stricture	High frequency	Low to high frequency
Pathological features	LPSP	GEL
Lesions of other organs	Sclerosing cholangitis	Inflammatory bowel disease (particularly UC)
	Sclerosing sialadenitis	
	Retroperitoneal fibrosis	

LPSP: lymphoplasmacytic sclerosing pancreatitis, GEL: granulocytic epithelial lesion, UC: ulcerative colitis

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
