# Peer review of "The Etiology of Pancreatic Manifestations in Patients with Inflammatory Bowel Disease"

_jcm, 2019, doi:10.3390/jcm8070916_

Round 1
Reviewer 1 Report
Good review. I would like the authors to emphasize that most of the studies are clearly association based studies. Authors should be careful to establish causality between different conditions.
Author Response
Reviewer 1
Good review. I would like the authors to emphasize that most of the studies are clearly association based studies. Authors should be careful to establish causality between different conditions.
→ Thank you for your comment. As you commented, various pancreatic manifestations with IBD patients are mutually associated and are not formed by a simple pathological condition. However, considering the total volume of this manuscript, it was difficult to describe them in detail. We added this in line 61-64 of page 2.
Reviewer 2 Report
In general, the manuscript entitled “The etiology of pancreatic manifestations in patients with inflammatory bowel disease” is well written and provides interesting data for the readership of the journal.
My only problem is that the manuscript is extraordinarily long (18 pages, of course mainly due to the long list of references), which in my view is too long. The manuscript could be considerably shortened (without harm) when all textbook information not related to the topic, which is pancreatitis in patients with IBD (but not different types of pancreatitis in general), are removed.
Unnecessary parts are for instance the first paragraphs of “acute pancreatitis” (2.2) and “chronic pancreatitis” (2.3.).
Just one minor issue: Amylase is abbreviated, but Lipase is not. Both should be dealt with in the same way.
Author Response
Reviewer 2
In general, the manuscript entitled “The etiology of pancreatic manifestations in patients with inflammatory bowel disease” is well written and provides interesting data for the readership of the journal.
My only problem is that the manuscript is extraordinarily long (18 pages, of course mainly due to the long list of references), which in my view is too long. The manuscript could be considerably shortened (without harm) when all textbook information not related to the topic, which is pancreatitis in patients with IBD (but not different types of pancreatitis in general), are removed. Unnecessary parts are for instance the first paragraphs of “acute pancreatitis” (2.2) and “chronic pancreatitis” (2.3.).
→ Thank you for your comment. As you commented, this manuscript covers various pancreatic manifestations in patients with IBD, however, we believe that any part is necessary to understand these pathological conditions.
Just one minor issue: Amylase is abbreviated, but Lipase is not. Both should be dealt with in the same way.
→ Thank you for your comment. According to your comment, we changed “AMY” to “amylase” in our manuscript.
Reviewer 3 Report
Iida et al. reviewed nicely the etiology of pancreatic manifestations in patients with inflammatory bowel disease. There are not many things that I want to ask to authors. However, i will be glad if they can add a small section where they can describe the role of microbiota in pancreatic manifestations of IBD patients. I am not really aware of the literature of microbiota and pancreatic manifestations in IBD. Having a couple of words showing we are with research on this subject can be beneficial for this review.
Author Response
Reviewer 3
Iida et al. reviewed nicely the etiology of pancreatic manifestations in patients with inflammatory bowel disease. There are not many things that I want to ask to authors. However, I will be glad if they can add a small section where they can describe the role of microbiota in pancreatic manifestations of IBD patients. I am not really aware of the literature of microbiota and pancreatic manifestations in IBD. Having a couple of words showing we are with research on this subject can be beneficial for this review.
→ Thank you for your comment. The relationship between microbiota and pancreatic disorders (acute or chronic pancreatitis, autoimmune pancreatitis...) has been investigated. However, as you commented, no papers on the role of microbiota in pancreatic manifestations of IBD patients have been reported. We believe future research is required to elucidate it.